# Trainee Therapists’ Perceptions of a Blended Intervention to Promote Resilience after a Natural Disaster: A Qualitative Case Study

**DOI:** 10.3390/jcm11154361

**Published:** 2022-07-27

**Authors:** Vera Békés, Geneviève Belleville, Jessica Lebel, Marie-Christine Ouellet, Zhaoyi Chen, Charles M. Morin, Nicolas Bergeron, Tavis S. Campbell, Sunita Ghosh, Stephane Bouchard, Stéphane Guay, Frank P. MacMaster

**Affiliations:** 1Ferkauf Graduate School, Yeshiva University, 1165 Morris Park Ave, The Bronx, NY 10461, USA; zchen6@mail.yu.edu; 2School of Psychology, Laval University, 2325 Rue de l’Université, Québec, QC G1V 0A6, Canada; genevieve.belleville@psy.ulaval.ca (G.B.); jessica.lebel@psy.ulaval.ca (J.L.); marie-christine.ouellet@psy.ulaval.ca (M.-C.O.); cmorin@psy.ulaval.ca (C.M.M.); 3Département de Psychiatrie, Centre Hospitalier de l’Université de Montréal, 1000 Rue St-Denis, Montréal, QC H2X 0C1, Canada; n.bergeron@umontreal.ca; 4Department of Psychiatry and Addiction, University of Montreal, 2900 Edouard Montpetit Blvd., Montreal, QC H3T 1J4, Canada; stephane.guay@umontreal.ca; 5Department of Psychology, University of Calgary, 2500 University Dr. NW, Calgary, AB T2N 1N4, Canada; t.s.campbell@ucalgary.ca; 6Faculty of Medicine and Dentistry, University of Alberta, 116 St & 85 Ave, Edmonton, AB T6G 2R3, Canada; sunita.ghosh@ualberta.ca; 7Département de Psychoéducation et de Psychologie, Université du Québec en Outaouais, CISSS de l’Outaouais, 283 Alexandre-Taché Blvd., Gatineau, QC J8X 3X7, Canada; stephane.bouchard@uqo.ca; 8School of Criminology, University of Montreal, 2900 Edouard Montpetit Blvd., Montreal, QC H3T 1J4, Canada; 9Addictions and Mental Health Strategic Clinical Network, Alberta Children’s Hospital, 28 Oki Drive NW, Calgary, AB T3B 6A8, Canada; fmacmast@ucalgary.ca

**Keywords:** telemental health, therapist, blended intervention, alliance, focus group, thematic analysis

## Abstract

Background: Natural disasters happen in an increased frequency, and telemental health interventions could offer easily accessible help to reduce mental health symptoms experienced by survivors. However, there are very few programs offered to natural disaster survivors, and no research exists on therapists’ experiences with providing blended interventions for natural disaster survivors. Aims: Our qualitative case study aims to describe psychologists’ experiences with an online, therapist-assisted blended intervention for survivors of the Fort McMurray wildfires in Alberta, Canada. Method: The RESILIENT intervention was developed in the frames of a randomized controlled trial to promote resilience after the Fort McMurray wildfires by providing survivors free access to a 12-module, therapist-assisted intervention, aiming to improve post-traumatic stress, insomnia, and depression symptoms. A focus group design was used to collect data from the therapists, and emerging common themes were identified by thematic analysis. Results: Therapists felt they could build strong alliances and communicate emotions and empathy effectively, although the lack of nonverbal cues posed some challenges. The intervention, according to participating therapists, was less suitable for participants in high-stress situations and in case of discrepancy between client expectations and the intervention content. Moreover, the therapists perceived specific interventions as easy-to-use or as more challenging based on their complexity and on the therapist support needed for executing them. Client engagement in the program emerged as an underlying theme that had fundamental impact on alliance, communication, and ultimately, treatment efficiency. Therapist training and supervision was perceived as crucial for the success of the program delivery. Conclusions: Our findings provided several implications for the optimalization of blended interventions for natural disaster survivors from our therapists’ perspective.

## 1. Introduction

Due to climate change, natural disasters and other extreme events have become more frequent and intense recently [1], and their devastating impact includes various mental health consequences for the survivors [2]. Previous studies estimated that up to forty percent of natural disaster survivors develop some type of mental health symptoms, including post-traumatic stress disorder, depression, and insomnia [3]. Psychotherapy has been found to be effective in the treatment of such problems in the context of natural disasters. For instance, after hurricanes Katrina and Sandy, cognitive-behavior therapy significantly reduced survivors’ symptom levels [4,5]. Even though it is essential to be able to provide psychological support after natural disasters, having access to sufficient mental health services at the location of the disaster is often a serious challenge.

Telemental health interventions have many advantages compared to in-person settings, including providing help for those who would not otherwise have access to mental health providers via in-person settings [6,7]. Among telemental health interventions, blended interventions are web-based interventions guided by a therapist that combine the advantages of asynchronous online programs and face-to-face, synchronous, personalized sessions with a therapist [8,9], which may be conducted either over the Internet or in person. Blended interventions have been suggested to have the most benefit by boosting treatment adherence, intensifying the learning process, personalizing and adjusting the intervention content to the client’s needs [9], as well as reducing associated therapist time and cost by providing a large proportion of the therapy content on the web to be utilized by the client [8]. Blended interventions appear to have the potential to make telemental health interventions suitable for more clients at lower therapist time and associated costs and may be a candidate for delivering telemental health interventions rapidly after natural disasters. Despite the many advantages of blended interventions over exclusively online, self-help modules, as well as over exclusively face-to-face sessions with a therapist, there are very few existing blended intervention programs providing help for natural disaster survivors.

Telemental health interventions have been underutilized mostly due to negative attitudes and concerns towards some of its aspects (see [10]) before the start of the COVID-19 pandemic. However, the advantages of telemental health interventions have become especially salient during the past months, when the COVID-19 pandemic and the related physical distancing restrictions forced therapists and clients to switch to remote therapy *en masse* [11]. Research suggests that therapists generally hold neutral or positive views of telemental health interventions [12,13,14,15]; however, providers are often reluctant to provide telemental health due to concerns and more negative views on it [16,17,18,19]. One common concern regards the ability to build therapeutic alliance [10,20,21,22]. Despite the fact that the quality of working alliance has been found to be excellent in videoconference (see review by [23]) and comparable to in-person therapies (e.g., [24,25]), providers are often worried that they would not be able to build strong rapport and develop alliance remotely [26]. Concerns and negative experiences, on the other hand, have been found to reduce the likelihood of intention to use telemental health in the future [26].

Another major concern regarding the providing of telemental health interventions is whether therapist and client can communicate emotions and empathy effectively remotely [6,27]. Despite research evidence that online channels do not influence emphatic accuracy [28], a common concern is that the online medium prevents the communication of empathy, warmth, and understanding [10]. Consequently, therapists are usually more favorably disposed toward videoconferencing as opposed to telephone or text therapy, as this modality enables them to observe physical cues and thus preserves an important element of in-person therapy [29].

Finally, therapists are also often concerned about the suitability of telemental health interventions to a certain type of clientele and situations [29,30]. Safety and legal concerns are common regarding interventions without physical proximity to the patient, and complex and severe client presentations, such as personality disorder, emotional instability, impulsivity, past suicide attempts, or any crisis situation are often perceived by clinicians as not suitable for telemental health settings [29].

However, we know that the therapists’ concerns and perceptions of telemental health vary. Recent studies showed that more previous experience with providing telemental health interventions and a larger teletherapy caseload were associated with more positive views on using telemental health [12,31,32]. Moreover, older therapists, and therapists with more clinical experience also reported more positive experiences with telemental health compared to their younger and less experienced counterparts [16,33].

### Fort McMurray Wildfires

On 1 May 2016 in Fort McMurray (Alberta), a major wildfire destroyed approximately 2400 homes and buildings and led to the massive displacement of 88,000 people, making it the most expensive natural disaster in Canadian history at the time. Many individuals faced direct or potential threats to their life or health, were separated from loved ones, and incurred significant losses. Families had to be displaced for several weeks and up to several months. For a report of the evacuees’ experiences three months and three years after the fires, see [34].

A year after the disaster, in a representative sample of 1510 evacuees, 38% reported symptom levels indicative of a probable diagnosis of either post-traumatic stress, major depression, insomnia, generalized anxiety, substance use disorder, or a combination of these disorders [35]. In order to provide support to the evacuees with these symptoms, our team of researchers and clinicians gathered an arsenal of well-evidenced CBT strategies into a 12-module online RESILIENT therapy program.

The RESILIENT platform Is a therapist-guided, online self-help treatment; its content was developed following a preliminary study of Fort McMurray residents who were evacuated during the fires, which showed that they had significant symptoms of PTSD, insomnia, and depression after the event [35].

The platform was hosted by Laval University, and the graduate psychology students were included as therapists in the program. A randomized control trial of 136 individuals who participated in the intervention showed that the RESILIENT program was successful and showed large effect sizes of symptom improvement in participants who completed at least half of the treatment. For further details on the RCT, see [36].

The present study aims to learn about the RESILIENT intervention from a different angle, by exploring trainee therapists’ first experiences with providing blended therapy for survivors of the Fort McMurray fires, in the context of the RESILIENT program. More specifically, our aims were twofold: (1) To explore therapists’ experiences with areas of common concerns described in the literature regarding telemental health services, such as the ability to build alliance online, communicate emotions and empathy, and suitability of remote interventions for all clients; (2) To explore therapists’ experiences with delivering the RESILIENT intervention, that is, their views on the intervention content and platform, as well as the training and supervision process.

We used a focus group design to learn about the therapists’ experiences. We selected a focus group design because, in contrast with individual interviews, focus groups allow for a discussion between group members, which typically widens the range of responses, brings up forgotten details, or triggers new ideas. In addition, due to the shared experience, the group dynamic often creates a sense of confirmation and of less inhibition in participants to freely share their views [37].

## 2. Method

### 2.1. The Resilient Intervention

The intervention was developed for a randomized controlled trial assessing the effectiveness of a blended intervention including online modules and synchronous therapist sessions either by phone or video conference for the evacuees of the Fort McMurray (Alberta, Canada) wildfires. Evacuees completed online symptom assessments, including PTSD (PTSD Symptoms Checklist, PCL-5; [38]), depression (Patient Health Questionnaire—Depression Subscale, PHQ-9, [39]), and insomnia measures (Insomnia Severity Index, ISI; [40]). Participants meeting inclusion criteria were offered free treatment. Inclusion criteria were: (1) significant post-traumatic stress symptoms (PCL ≥ 23) or (2) some post-traumatic stress symptoms (PCL ≥ 10) and at least mild depressive symptoms (PHQ-9 ≥ 5) and/or at least subclinical insomnia symptoms (ISI ≥ 8).

The intervention consisted of a therapist-assisted online cognitive-behavior therapy focusing on post-traumatic stress, sleep, and mood. It comprised 12 sessions with modules of psychoeducation about PTSD, sleep and depression, prolonged exposure to avoided situations and memories, sleep management strategies (restriction of time in bed, sleep hygiene education, nightmare image rehearsal), behavioral activation, breathing and mindfulness exercises, cognitive restructuring, and relapse prevention. Participants had to work on each week’s session online individually by reading educational material, reflecting on their own experiences by answering questions, planning their homework exercises, and filling out online journals about sleeping, breathing exercises, exposure, and behavior activation. They met a therapist via either videoconference (Skype) or phone, according to the participants’ preference, after completing each session to discuss the given topic and review their online work as well as the homework (exercises in-between sessions). Meetings with the therapists were 30 min long, and usually weekly sessions, depending on the participant’s online progress. Preliminary results have been previously published in French [41].

### 2.2. Participating Therapists

Seven clinical psychology graduate students, that is, all the therapists in the program, participated as therapists in the program. All therapists were female with a mean age of 25.86 (range: 22–30, SD = 3.08). None of the therapists had previous experience with remote therapy. Therapists had about 2 years of in-person clinical experience (mean = 2.13 years; range: 0–4.5, SD = 1.67), and only one therapist was beginner. Altogether, 69 clients were invited to participate in the program and were assigned to a therapist; each therapist had 9.75 clients assigned to them on average (range: 3–17, SD = 3.99).

### 2.3. Procedure and Data Analysis

Two focus groups were conducted to collect data from each participating therapist shortly after the termination of the intervention. Each focus group lasted for 120 min. All 7 therapists provided data about their experiences by participating in the two focus groups (*n* = 3 and *n* = 6, two therapists participated in both). The focus groups used a semi-structured interview guide with open questions. Questions focused on general impressions regarding the intervention, perceived challenges for the therapists and the clients, and more specific questions regarding experiences with phone and video conferencing with the clients (such as developing alliance and feeling empathy), the online platform, the content of the intervention, and finally, suggestions for improving the intervention. The focus group interviews were audio recorded and later transcribed. Given the bilingual milieu, where all participants used both English and French in professional and daily communication, the therapists responded in either of these languages within the same group, according to their preferences, or switched from one to the other language as they would naturally do outside of the focus group setting. The possibility to use either English or French allowed participants to express themselves in the language they felt more comfortable with, while not risking hindering the process of understanding. The transcription kept the original languages, and quotes in French were translated to English for the present study.

Two graduate clinical psychology students (who did not participate in the study) conducted thematic analysis on the transcribed data on the guidelines provided by Braun and Clarke (2006). The qualitative analysis included the following phases: (1) generating initial codes, (2) identifying emerging common themes, (3) reviewing the themes, arranging the themes under a priori established categories (building alliance, communicating emotions and feeling empathy, suitability of intervention) as well as defining emerging categories (platform, intervention content, training, and supervision), and (4) finally, refining the definition, specifics of each theme, as well as the thematic map). Illustrative quotes were selected and translated from French to English. Ethical approval for the study was obtained from the Laval University Institutional Review Board, and participants provided informed consent.

## 3. Results

The themes were categorized under general concerns regarding alliance, communicating emotions and empathy, and suitability (Table 1), and specific issues, including intervention content, platform, and training and supervision (Table 2).

### 3.1. Alliance and Client Engagement

Despite previous apprehensions, all therapists had generally positive experiences with establishing and maintaining alliance with clients via phone calls and video conferencing (7/7); as one of them described “*It was more rich than I thought, since there was a screen between us, it was interesting to see that I was able to bond with the [clients] even though they were far away, so yes, that surprised me*.” (P1). Most therapists felt that video sessions were more helpful for feeling closer to clients (5/7), and they had positive experiences with phone sessions in general, although some signalled that it took more time to build rapport over the phone (1/7).

Client engagement emerged as a central theme affecting alliance: the therapists felt that they worked well with motivated clients, whereas client non-compliance negatively affected the alliance (7/7). In reaction to issues with engagement, therapists realized that providing regular encouragement, motivation, and checking in were needed to ensure that clients adhered to the intervention (3/7). Moreover, according to the program protocol, in order to maintain client engagement, therapists had to regularly reach out to clients when they had not heard from them. This was frustrating for most therapists and felt like needing to constantly reach out to clients (4/7), which sometimes even felt like they were harassing the clients (2/7). The therapists’ impression was that the frequent reaching out process appeared to be frustrating for both them and the clients, and most of the time it did not reach its goal of increasing engagement but most probably negatively affected their alliance.

### 3.2. Communicating Emotions and Empathy

Overall, therapists felt they could adequately sense and communicate emotions and empathy both via video conferencing and phone (7/7). They felt that even when only using audio (phone) and not video, they were able to recognize clients’ emotions, understand their perspectives, and empathize with their traumatic experiences (4/7). One therapist reported that phone sessions, compared to face-to-face, made her less stressed and self-conscious, and consequently more focused with the clients, allowing her to develop better empathy.

Despite being comparable in many aspects, video call sessions were viewed as more favorable than phone sessions in general for various reasons. First, the therapists had better experiences with video sessions with regards to communicating empathy, as they felt a “better presence” with clients (3/7). Second, they also felt that it was harder to assess the clients’ mental state via phone due to a lack of information about their facial expression (4/7). Third, lacking access to non-verbal cues also limited the therapists’ ability to keep the sessions focused on the given topics and not to divert into others (3/7). Some therapists voiced concerns that there might be more distractions around clients during phone session that are hard for the therapist to detect (1/7), and thus sometimes they wondered what the clients were doing during the sessions (2/7).

Despite the overall preference for video calls over phone calls, therapists also experienced its limitations. For example, access to the clients’ nonverbal cues were limited to facial expressions and diminished the therapist’s ability to transmit nonverbal signs (7/7). Moreover, even though therapists could read and communicate emotions via video call in general, the angle, distance, and resolution of the camera set limits to what the therapists could observe, limiting their access to some nonverbal or social cues (5/7).

### 3.3. Suitability and Client Engagement

According to the therapists, the single most important factor in the success of the intervention was whether the client was motivated and engaged. In turn, client engagement greatly differed based on certain client and situation characteristics (7/7). Therapists felt that client engagement was often polarized: clients were either motivated and active from the beginning until the end, or were less engaged, non-responsive to emails, missing sessions, and the therapists’ efforts to motivate them had a very limited impact on changing this: “*Either they were engaged and following every week or it was difficult to engage them and they would do one to three sessions and then it was very difficult, and it took two or three weeks to do another one and they would always postpone it. So, I think it was the main thing with my participants*.” (P6) According to the therapists, clients were less engaged if they did not feel that their symptoms were targeted (4/7), for example, for clients who did not have avoidance symptoms, a central focus of the intervention appeared unrelatable.

Therapists also proposed that the presence of severe symptoms or other ongoing life stressors might have contributed to clients’ non-adherence and dropout (2/7): “*I only had one participant that dropped out, and for her I think that her symptoms were really severe and for her to do it by herself on the computer, it was just too much, she told me that the first or second sessions she started having nightmares, felt a lot of anxiety. She had flashbacks again, so I think that’s why, she told me that, and she wasn’t sure if she wanted to continue or not. And after that she emailed me back and she never answered, so she dropped out after that.*” (P5)

Furthermore, mismatch between client expectation and the offered intervention negatively affected client engagement. Most of the time, the client did not expect that the intervention would take so much time, effort, and emotional investment (7/7), “*They didn’t know what they were getting into. Even though we told them in the beginning that it was a self-based intervention, I don’t think that they knew exactly what it was (…) I don’t think that they expected so much work.”* (P2)

### 3.4. Intervention Content

The therapists found the intervention was well-made, the content was clear, well-organized, easy to understand, and that the modules were organized in the right order (7/7); however, the utilization of the exercises mostly depended on client engagement (7/7). According to therapists, engaged clients claimed that this intervention made a difference in their lives (2/7) and especially appreciated the therapist sessions in addition to the online modules (2/7). The utilization of the exercises also depended on the specific needs and symptoms of the clients, and therapists made efforts to repeat helpful exercises and skip less relevant ones in order to adapt to the client’s needs (4/7). “*It was really different from one person to another. I feel like all of my participants had their one or two that they really liked and kept during all the program and I don’t believe that any of the intervention wasn’t useful, I think it just depends on the fit with the person. Because they are all in themselves relevant, it just depends on the match*.” (P1) The clients sometimes focused on a sole exercise, which still led to significant improvements: “*The very use for she was the pleasant activities [behavior activation] and so this is something she really did regularly. So yes, it was helpful and she had issues if I may say, but at the end she had no issues with sleep, and all the things so at the very end she only had to complete the pleasant activities and that was it*.” (P1)

The therapists felt that exposure was difficult to do alone (2/7) and in-person assistance would have been helpful for clients (1/7). The insomnia element of the intervention was very helpful for those who had symptoms (5/7), although the online sleep diary was difficult to use (6/7). Behavior activation exercises were popular and helpful in general (4/7). The cognitive restructuring exercise, although the therapists thought it was important, was hard to work with in the blended therapy’s frames, as clients would have needed more support to recognize their own maladaptive thoughts and finding alternatives (4/7). The most utilized exercises were the insomnia, behavior activation, and diaphragmatic breathing interventions (4/7); less utilized exercises included nightmare imagery rehearsal, given that very few clients had nightmares (7/7). Utilization of the online tools, for example sleep diary, and anxiety monitoring before and after interventions, greatly varied, and the majority of clients stopped using them after a few times (6/7). For more detailed description of experiences with the specific exercises see Table 2.

#### 3.4.1. Intervention Platform

Therapists agreed that the platform was clear and easy-to-use, visually pleasing, and inviting (7/7), and most therapist did not experience technical problems. However, clients sometimes forgot to enter their exercise data (2/2) and a technical glitch that resulted in losing client data was very frustrating for clients (1/7).

#### 3.4.2. Training and Supervision

Therapists felt they needed to work on getting prepared for providing the intervention by reading and becoming familiar with the content, exercises, online tools, and learnt a lot about therapy principles and practice (5/7) while receiving a meaningful training experience (1/7). The therapists all felt that the supervision was helpful in providing ideas and a new perspective (4/7), as well as support and frames for approaching complicated situations (2/7) and setting boundaries (1/7).

### 3.5. Underlying Processes: The Role of Client Engagement

In summary of the results, the following underlying process emerged. Client engagement appears to have a fundamental role in the success of the treatment, with regards to developing strong alliance, provide effective communication, and lead to significant symptom improvement. Client engagement, on the other hand, largely depends on the suitability of the intervention for the client. For clients with severe symptoms or symptom profiles that do not match the intervention’s focus, or for clients with differing expectations, the intervention proved to be less suitable, leading to a lack of client engagement, which negatively impacted the working alliance and communication between client and therapist and, ultimately, symptom improvement (Figure 1). Moreover, besides suitability to client expectations and symptoms, the utilization and helpfulness of specific exercises greatly varied depending on their suitability to the blended intervention format that heavily relied on asynchronous communication. Some exercises worked well without synchronous therapist support; for example, behavior activation or diaphragmatic breathing, whereas others were challenging to execute without in-person or more synchronous remote support (such as exposure exercise, cognitive restructuring).

## 4. Discussion

The present study aimed to explore therapists’ experiences with providing telemental health in the context of a blended intervention for survivors of a natural disaster. We found that therapists had an overall positive experience with the ability to build alliance, communicate emotions, and empathize with clients via both phone and video sessions, while they identified special advantages and challenges within each domain. Client engagement emerged as a central underlying theme that was perceived as having a fundamental impact on both the process and the outcome of the intervention. This result is in line with a subsequent quantitative analysis of usage data of by the same participants, where Label and colleagues [42] recently found in the same sample of clients that treatment efficacy (reduction in post-traumatic stress, depression, and insomnia symptoms) was related to the number of modules accessed by the client, which, in turn, was predicted by previous engagement (e.g., number of words entered) in preceding modules.

In our study, client engagement was perceived as depending on (1) the match between the client’s symptom profile and the intervention content, (2) symptom severity (overwhelming, severe symptoms often led to disengagement and eventual dropout), and (3) the match between client expectations regarding the program and the actual intervention. Moreover, in our trainee therapists’ view, utilization of specific exercises greatly varied based on the difficulty to execute them without direct and synchronous therapist support within the blended intervention paradigm.

### 4.1. Alliance

Similar to previous findings (e.g., [26]), trainee therapists in the present study had concerns about the ability to build alliance prior to starting the intervention and were pleasantly surprised by the ease of creating bonds remotely. This finding supports the notion that once therapists are effectively engaged in delivering online intervention, their experiences are usually positive with alliance building [10,11]. While creating bonds proved to be easier than expected, trainee therapists still faced various challenges regarding specific aspects of working with clients. They felt that the ability to build alliance greatly depended on the clients’ engagement to the program (as seen in previous studies as well, e.g., [43]), and maintaining clients’ motivation posed a fundamental challenge to therapists. Based on previous recommendations (e.g., [44]) the study protocol required therapists to reach out to inactive clients regularly in order to maintain engagement; however, this method appeared to be counterproductive, resulted in therapists feeling like they were “chasing the clients”, and posed a strain on the alliance on both sides. Eventually, our therapists found different ways to keep clients engaged, tailor-made for each client, which they perceived as more effective.

Moreover, in contrast with earlier findings where the aspect of alliance regarding agreeing on the goals and tasks was found to be higher in online therapies [45], for our trainee therapists this posed a challenge. Since the modules, exercises, and their order had already been pre-established, the therapists had little space to adjust them to the needs of the actual clients, which frequently resulted in a mismatch between client expectations and the delivered intervention. Moreover, since clients were recruited by the RESILIENT program by offering a free intervention for those with above-threshold symptoms, many clients may not have had a strong motivation to begin with, and when faced with the time and emotional investment required by the program, became even less motivated.

Problems with high degrees of non-adherence are common in online interventions [46], and blended interventions’ synchronous therapist support component are aimed to address this; however, our results suggest that, in itself, this might not be enough. The clients’ specific needs could be better met by making the intervention content and sequence more flexible; this would increase client and therapist agreement on tasks and goals, and consequently, have a positive impact on client engagement. For example, implementing a preliminary phase at the beginning of the intervention where the client’s symptoms are assessed, and matching interventions be selected and arranged in the desired order may be helpful. This phase could be either conducted by a therapist or by an online algorithm, or by a combination of both where the therapist is able to adjust the algorithm’s recommendations based on a collaborative discussion with the client.

### 4.2. Communicating Emotions and Empathy

Similar to experiences with alliance, trainee therapists were pleasantly surprised by the ease of communicating and feeling empathy both over the phone and in video calls. Video sessions were perceived as preferable to phone due to the availability of visual information and thus the ability to sense the clients’ emotional states and understand their mind states. At the same time, even video calls filtered out the perception of certain nonverbal clues, a common issue that has been raised in previous studies. To address this, Grondin et al. [47] suggested that techniques like exaggeration of nonverbal behaviors and verbal clarification of the client’s affective state can facilitate the empathic phenomenon. Moreover, simple technical adjustment in the video camera placement and settings can also increase connection and the perception of nonverbal cues. For example, setting the camera angle in a way that enables eye contact, and selecting the appropriate camera frame (zooming in and out to show the usual head-to-chest versus larger, head-to-waist frame) allows the perception of nonverbal cues while also maintains a sense of psychological connection [48]. Future studies need to explore the utility of these techniques in telemental health.

### 4.3. Suitability

Previous studies suggested that online interventions might be less suitable for certain clients, for example, those with severe or complex psychopathology and in crisis situations [29]. In our study, although some of the trainee therapists mentioned symptom severity contributing to unsuitability and non-adherence, the relevance of the intervention content to the client appeared to play a more important role in the client–intervention match. As for alliance, our results regarding suitability indicate the need for developing more personalized treatments in telemental health, and specifically, in blended interventions, instead of utilizing a one-size-fits-all approach [49]. Bettering the fit between client needs and expectations and provided content could improve client engagement and adherence, which has been a major challenge in our current as well as in previous studies.

### 4.4. Helpful Exercises

Therapists perceived certain interventions as better fits, while others as less good fits for blended interventions. Interventions that need strong emotion regulation skills (e.g., in vivo exposure exercises) or ability to critically engage with one’s thoughts and beliefs (cognitive restructuring) may be challenging for many clients in a remote setting that heavily relies on independent client work. Providing more synchronous therapist contact, i.e., more frequent sessions, or simplifying these exercises from the given client’s treatment protocol could address this problem. Other exercises that need little therapist support but are highly successful (e.g., behavior activation) could also be included in exclusively web-based intervention protocols.

### 4.5. Professional Support

Training and supervisor support before the preparation and throughout the intervention was perceived as crucial for the trainee therapists, who were young graduate students with relatively little clinical and no telemental health experience. They reported that training and first-hand experience with telemental health positively impacted their attitudes in our study, similarly to earlier findings [11,50]. Since experience with telemental health is also associated with more therapist confidence regarding the ability to build alliance, read emotions, and be emphatic with clients online [26], in order to promote the utilization of telemental health among providers, providing training and ongoing professional support for trainee and novice therapists would be crucial.

### 4.6. Limitations

As all studies, ours had its limitations as well. First, we had a relatively small sample of participant therapists; however, this is not unusual in qualitative studies. Qualitative inquiry typically uses smaller sample sizes (sometimes even a single participant, [51]) in order to explore a limited number of participants’ subjective experiences in depth, in contrast with quantitative studies, where a larger random sample may better represent the views of a general population [52]. Based on this notion, we included a small number of therapists with the specific experience of participating in the RESILIENCE program, instead of a larger sample of therapists with potential experience in other programs. In addition, for a case study like ours, a focus group design with a relatively small number of participants is recommended, where the participants who are all familiar with the case (i.e., the RESILIENT intervention] are able to share experiences and generate ideas together [53]. Therefore, the fact that we could include all the participating therapists in our focus groups could be in fact considered as a strength of the study.

Second, the trainee therapists’ experiences with providing telemental health within the context of the RESILIENT intervention in rural Canada might not be transferable to other circumstances. Our trainee therapists were all junior with little or no previous therapy experience, and despite the common view that younger people feel more comfortable using technology, earlier research found that more junior therapists with less clinical experience tend to have *less* positive experiences with providing telemental health compared to their older and more experienced counterparts [16,33]. Therefore, our therapists’ perceptions of the intervention may differ from what older and more experienced therapists would experience under the same circumstances. Moreover, our participants, the specific event of the Fort McMurray fires, and the circumstances in our study were also unique, which may further limit the transferability of the findings.

Furthermore, our therapists within the study had some differences, such as their previous clinical experience and present caseload, which might have also impacted their experiences [35]. However, this case study’s findings regarding therapist concerns and initial first-hand experiences with blended interventions for natural disaster survivors may still be helpful when preparing for new interventions under different circumstances.

## 5. Conclusions

The therapists in our case study perceived the blended intervention as a significant mental health tool for survivors of a natural catastrophe. The perceived success of this model encourages the implementation of similar blended interventions for survivors of other natural catastrophes in remote areas, or where psychological help is not readily available, as well as for survivors of other type of traumas who are reluctant to seek help in person, for example, sexual assault victims. However, based on our therapists’ experiences, in order to improve the efficiency of such interventions, personalization of the treatment content and sequence, as well as proportion of therapist sessions and web-based content based on client needs, is recommended. Furthermore, providing theoretical and skills-based training in telemental health is recommended to improve the quality of online interventions in crucial areas, such as providing relevant content, building alliance, and specifics of online communication with clients. Our hope is that the accumulating knowledge of the specificities of online interventions for natural disaster survivors will not only inform the development of blended interventions for natural disaster survivors in the future but may also be incorporated in the training of future providers.

## Figures and Tables

**Figure 1 jcm-11-04361-f001:**
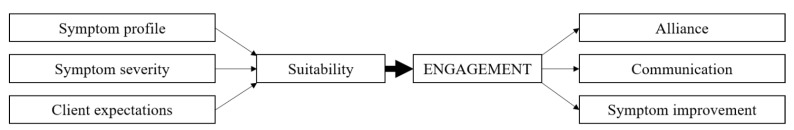
Thematic map of therapists’ experiences with alliance, communicating emotions and empathy, and suitability. *Note*. Client engagement was crucial for the success of the treatment, as it tended to lead to stronger alliance, effective communication, and significant symptom improvement. Client engagement, on the other hand, largely depended on the suitability of the intervention for the client. The intervention was less suitable for clients with symptom profiles that did not match the intervention’s focus, for clients with severe symptoms, and for clients whose expectations did not match the intervention. Problems with suitability lead to a lack of client engagement, which negatively impacted the working alliance and communication between client-therapist and, ultimately, symptom improvement.

**Table 1 jcm-11-04361-t001:** Therapist Experiences with Alliance, Communicating Emotions and Empathy, and Suitability.

Description	N	Supporting Quotes
** *Alliance in General* **		
Therapists were able to develop strong alliance both in phone and video sessions	7	“One can create a good therapeutic alliance with the client both by phone or Skype. Active listening and validation of their symptoms make clients already more comfortable with disclosing their feelings.” * (P2)
Video helped feeling closer to clients more than phone sessions	5	“I found that over video it was easier (…) to feel that you were really there with the client.” * (P2)
Phone sessions felt similar to face-to-face	2	“I didn’t feel there was a difference to face-to-face, sometimes it [the distance] even helped me personally.” * (P1)
Communicating via phone didn’t have negative impact on the relationship	2	“I felt that I had a good therapeutic relationship with them, for example in the end it was difficult for them to say “bye,” so there was a relationship that had been built.” * (P5)
Communicating via phone required longer time to build rapport	1	“I think in the beginning it takes a little bit more time to develop the therapeutic alliance, because you’re picking up the phone and all of a sudden you are expected to divulge your whole story to at a random person on the phone.” (P4)
** *Alliance and Client Engagement* **		
Therapists worked well with those who were motivated and adherent to the program	7	“I found that I had as much empathy as when I saw a client in person. The only thing that affected my ability to be empathetic was with one particular participant. It had been twice that I called her at the time of the appointment and she did not answer and she always postponed the appointment so I was irritated by that so I tell myself that maybe it has affected my ability to be truly empathetic.” * (P6)
Reaching out to clients led to frustration on both therapist and client sides and negatively affected alliance	4	“Like I was a mother running after her children pining at them, because I knew it would help them, they just weren’t committed to the intervention whatsoever, it was just frustrating the whole time.” (P5).
Clients are less engaged if the specific intervention does not match their symptoms	4	“The participants I followed didn’t really have the classic symptoms of PTSD necessarily so maybe it was more difficult for them to recognize themselves in like specific categories.” (P2)
Regular encouragement needed to increase engagement	3	“Sometimes I went through their answers and gave them little reinforcements. I would tell them that this was a good idea or I would encourage them to go a bit further in their responses.” (P2)
Mismatch between client expectation and intervention content led to frictions between therapist and client	3	“They commit to the program, but they don’t really know in what they got involved. When they discover that it’s a serious program and that we meet with them every week… I think it might take too much of their time.“ * (P3)
Personal contact with therapist improved client engagement	3	“I think (…) they really appreciated the opportunity to talk things over.” (P6)
Reaching out feels like harassing the client if they do not respond	2	“At some point I was feeling like I was harassing this client who hasn’t been answering me.”
Challenging to maintain alliance when sessions are too far away in time	1	“When it was difficult to schedule an appointment, it got stretched out. I had one client who (…) took really long. It was more difficult, because the alliance was not there anymore (…), it was with him that I had to tell him that I had a life too.” * (P2)
** *Communicating Emotions and Empathy* **		
Therapists were able to feel and communicate emotions and empathy	7	“I found that I had just as much empathy as with a client in person.” * (P6)
Harder to transmit nonverbal signs	7	“It’s a little bit more artificial to say « ok, like, now it’s the end » instead of just giving social signs that we are close to the end.“ (P1)
Nonverbal and social cues were limited even in video conferencing	5	“For example if a person felt uncomfortable and was playing with his pencil, I did not see it on his face as it was too close [to the camera], so maybe there were things I missed.” (P5)
Therapists were able to recognize clients’ emotions, understand their perspectives, and empathize with their traumatic experiences	4	“I had some doubts [about remote sessions] but speaking to the victims and feeling their emotions through the phone made me want to participate in this project even more; knowing that they really experienced difficult things that affected them a lot, and all that over the phone.” * (P2)
Video is better in assessing clients’ mental state due to visual information	4	“I had one participant over the phone and obviously, zero nonverbal, it isn’t there, at one point she became emotional, and she was crying, but I didn’t know she was crying because it was silent. So it was kind of not how you would intervene if it were on video, because you would see that the client was emotional and I would have given her space and let her live these emotions and instead I was like « hello, you still there »” (P5)
Hard to sense the meaning of silences on the phone	3	“You can have silence on [video call], but on the phone you don’t know what the person’s doing. So I feel like they have to fill more the blank on the phone than when it’s face to face.” (P1).
Hard to focus, clients tend to chat about other things via phone	3	I think it was harder on the phone for … to be focused on the content, I think over the phone it was easier to talk about something else. (P2)
Video is better to express empathy	3	“I was wondering, did they feel the non-verbal empathy that we try to show when we are with the person? (…) When they start to cry for example. I felt our silence afterwards was supportive, but I am not sure it worked.” * (P1)
Hard to get a sense of the clients’ actual surroundings, activity on the phone	2	“You had no idea what they were doing, she could have been watching TV, she could have been doing anything. And well, on Skype I can see, I just felt like it was a lot better on Skype than on the phone.” (P5)
Easier to focus on the client and feel empathy on the phone due to decreased therapist anxiety	1	“I may have little performance anxiety but as the person was far away it allowed me to focus on the person, rather than on what I looked like, right? (…) I was more focused on what the person was saying to me, the distance helped me in terms of empathy.” * (P1)
** *Suitability* **		
Clients’ presentation of symptoms affects their intervention utilization	7	“The participants I followed didn’t really have the classic symptoms of PTSD necessarily, so maybe it was more difficult for them to recognize themselves in like specific categories or specific boxes [in the online modules], which is why it was more difficult for them to understand [the exercise]” (P2)
Clients differ in their commitment to the intervention	7	“I had two or three participants that … in a week did sessions 2 to 9 or 2 to 6. I imagine it depends also how committed they are and if they do all the exercises.” * (P3)
Lack of accurate expectation of workload in the intervention affected clients’ engagement	7	“Some people have never been in therapy, so they don’t know what it’s like and when they are recruited, we say « oh you want this treatment for insomnia, etc. » and I don’t think they expected it to be this big, this demanding. That they would have to monitor many things, be accountable.” (P6)
More symptoms/stress leading to dropout	2	“I feel like maybe because they have severe symptoms and they have a lot of things going on in their life, (…) they were telling themselves that it was more important than the intervention, so they were putting it on the back burner. It was not their priority so I think it explains a lot of the drop-outs.”

*Note*. * = Translation from the original French.

**Table 2 jcm-11-04361-t002:** Therapist Experiences with the Content, Platform, and with Training and Supervision.

Description	N	Supporting Quotes
**Intervention Content**		
** *Exposure exercise* **		
Less utilized due to lack of avoidance symptoms	2	“Many of my participants didn’t do any exposure because they weren’t avoiding anything related to the fires.” (P6)
Challenging as it provokes anxiety	2	“The exposure part of the modules, this was challenging, some of them told me that it was hard for them to do [experience] it all over again.” (P2)
Clients did not believe it would be useful	2	“It was a challenge for me to try to explain how it would be useful to them if they didn’t think it was useful themselves.” (P5)
Clients found it helpful	1	“I had two people who found it really useful, it was really really useful, so I think it depends on the person. And they were very surprised at how it helped them.” (P1)
Assistance would be helpful	1	“I think sometimes the exposure was (…) hard for them to do on their own. Sometimes I wanted them to include somebody that they are comfortable with, but sometimes it was hard. So I think that [it would be helpful] if there was a therapist with them.” (P2)
** *Sleep management (Sleep window, Sleep diary, Nightmare imagery rehearsal exercises)* **		
Sleep diary was hard to follow	6	“I wanted to use it more, but I think sometimes the participants didn’t put the right sessions when they filled the entries, so it wasn’t always very accurate. So sometimes I wanted to come back to the [sleep] efficiency, but I couldn’t. (…) It’s kind of complicated I think.” (P2)
Helpful for those who used it	5	“I only remember a few people who really used the sleep diary and for whom there was no trouble and it was very helpful but not a lot of people did it.” (P2)
Sleep diary was helpful in improving sleep	3	“I found the most useful for my participants was the sleep diary for the people that had sleep problems who were really working on improving their sleep.” (P4)
Sleep window is difficult to do and requires therapist’s explanation	1	“The only section that I found a little bit more difficult, (…) was about the sleep window, just because there’s tons of noting, there’s monitoring, there’s adding fifteen minutes here, so it’s likI. That was really the only aspect that I felt needed to be explained more in detail to clients (P4)
Sleep diary was not utilized as it felt irrelevant	1	“None of my participants did it and they found it to be really irrelevant.” (P5)
Nightmare imagery rehearsal was less utilized due to lack of nightmares	7	“I think only one of them [client] used it because the other ones didn’t have nightmares.” (P6)
** *Behavior Activation (Pleasant activities exercise)* **		
Utilization and adherence varied	5	“Either they use it a lot or they use it for like the first week and then they stop.” (P6)
Adaptable, useful, and helped most clients	4	“I think pleasant activities were the most used. They were used by all of my participants. It was something that they can all recognize themselves.” (P2)
* **Cognitive Restructuring Exercise (Unhelpful thinking styles)** *		
Difficult to do by themselves	4	“I think for some of them [used it], but for others it wasn’t natural, some of them don’t have access to their other alternative [thought].” (P2)
Challenging to explain	4	“Some participants couldn’t recognize their unhelpful thinking styles, so what can you do at that point? You know, you can do nothing if they don’t see.” (P1)
** *Diaphragmatic Breathing and Mindfulness (Calm breathing, Mindfulness mediation exercises)* **		
Diaphragmatic breathing exercise was most helpful	3	“It was the breathing exercises […] that was a really important one for most of my participants” (P7)
Meditation may work better with auditory format	2	“I wonder if it could be an idea to do a recording instead of a text. Because meditation is more of an auditive side than visual.” (P6)
Mindfulness meditation was helpful	1	“I feel like the mindfulness exercises really helped a lot of my participants specifically. With one participant she took it and she started journaling with it between the sessions and she told me at the end that it’s something she’s going to continue to do for the long term because it really helped her.” (P5)
**Platform**		
Well-functioning and user-friendly	7	“I thought the platform was great as well. It was visually pleasing, inviting for participants, interactive because there were many things to click on.” (P2)
Clients forget to record their exercises	2	I think it was more time consuming or they would do things and they said ‘’Oh! I need to write it down’’ and then they would forget to write it down once they actually got access to the platform. This is why I suggested keeping a paper journal or an agenda.” (P2)
Technical glitches are very frustrating for clients	1	“They would get frustrated or annoyed if they had completed their sleep diary for an entire week and then it didn’t save.” (P4)
**Therapists Training and Supervision**		
Therapists needed to get familiar with the content and tools	5	“I felt like I learned a lot since it was only my first year here so I learned a lot on intervention and on CBT in general so it’s really enriching for me.” (P5)
Supervision was helpful, available for questions	4	“I think that, honestly, it was very useful, very helpful. If I had any questions, I could address them to her [supervisor] directly. (…) She helped me deal with that or using it appropriately or how I should be intervening other ways.” (P2)
Supervision needed in complicated situations	2	“I found it good… Participants sometimes had more complicated situations. So at least we had a different way of thinking about the issue. She [supervisor] could give us clues on what to do.” * (P3)
Meaningful training experience for therapists	1	“In general, I felt like it was a really rich experience for me as well, I thought the program was really complete and really rich, I felt like I learned a lot as well since it was only my first year here, so I learned a lot on intervention and on CBT in general so it’s really enriching for me.” (P5)
Supervision helped keeping frames and communicating expectations	1	“Most of the time, we just postponed it [the session with a participant] to the next week and at some point, I asked [the supervisor] for advice and she told me to ask them to cancel 24 h in advance and be more… not strict but have more like a frame on how to do things.” (P6)

*Note*. * = Translation from the original French.

## Data Availability

Not applicable.

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
