# Peer review of "Trainee Therapists’ Perceptions of a Blended Intervention to Promote Resilience after a Natural Disaster: A Qualitative Case Study"

_jcm, 2022, doi:10.3390/jcm11154361_

Round 1

Reviewer 1 Report

This study evaluates psychologists’ experiences with an online, therapist-assisted blended intervention, for survivors of a natural disaster in Alberta Canada. They found that the intervention was suitable and easy to use despite its limited efficacy for participants experiencing higher levels of stress and those less engaged in the program. It appears that client engagement played a great role in the creation of therapeutic alliance and treatment efficiency. The issue is relevant as more flexible programs are needed to deliver care in hard-to-reach communities in need of support. However, the value of the present work is limited by the fact that efficacy of the intervention program is based on therapists' perspective only and above all the small sample (n=7) considered in this study.

I found it confusing the use of many terms such as “telemental health”, “web-based program”, “online therapist assisted intervention”, “blended” in an almost interchangeable way in the text. I would urge the authors to be mindful  of the terminology used and to try to unify if as much as possible to avoid confusion.

Reviewer 2 Report

Dear Authors

I find your paper and research interesting and suitable for the journal. Still, some things should be from my perspective clarified and explained:

  1. You call the intervention blended that is normally understood in the literature as combination of online and offline measures, which is to my understanding not the case there. So what is the context of this term in your text?
  2. What is the reason for using focus groups? What is the added value off the interaction – in comparison to individual in-depth interviews? Particularly in the context of using two languages during the sessions? I do not fully understand this methodological choice.
  3. I would definitively remove percentages from the table (they are not reasonable and misleading when N=7)
  4. I suggest not using “social distancing” as criticized by WHO (we needed only physical distancing with social connection – even when only online during the pandemic).
  5. Please discuss limitations of the study – eg. Age of respondents who as younger may be more enthusiastic for online interventions.

Best regards,

Reviewer 3 Report

Dear Authors,

thank you very much for the manuscript. It addresses a very relevant issue, considering that due to climate change the frequency and intensity of extrem events (I would not call them "natural disasters"), including wildfires, will probably increase in the future. I liked the manuscript very much, but there are a few points to be clarified and improved:

1. How many evacuees have been assisted by the participating therapists? All therapists assisted the same number of evacuees? Differences on the number of assisted evacuees could have had eventually an influence on therapists' perception (?) I suggest/recommend to clarify this point;

 2. Since perception is not an entirely objective phenomenon depending on both the observer and the observation context, and although item 4.6 made a quick comment on this (“Our trainee therapists were all junior and thus their perception may differ from more senior therapists’ perceptions”), I missed a more in-depth discussion on this topic. Perhaps include a topic about this in the Introduction?

 3. I don´t think Figure 1 is sufficiently clear, it is not “self-explaining”. Could it be improved?

I hope these comments may help you to improve the manuscript. I wish you all the best!

Round 2

Reviewer 1 Report

I didn't find any feedback on part of authors to my comments. I reiterate that the greatest limitation of the study rests on the small sample size (n=7).

Author Response

Thank you for your review, we have responded to it on May 9 and uploaded the response on this site, I can see it under "report notes." I am uploading the response again, please see attachment.

Reviewer 3 Report

Dear Authors,

thank you for the revised version of the manuscript.  I consider that my requests have been granted.

Author Response

Thank you for reviewing our manuscript and for the helpful suggestions.

This manuscript is a resubmission of an earlier submission. The following is a list of the peer review reports and author responses from that submission.